# *In Vivo* Biomechanical Measurements of the Cornea

**DOI:** 10.3390/bioengineering10010120

**Published:** 2023-01-15

**Authors:** Fanshu Li, Kehao Wang, Ziyuan Liu

**Affiliations:** 1Beijing Key Laboratory of Restoration of Damaged Ocular Nerve, Health Science Center, Peking University, Beijing 100191, China; 2Beijing Advanced Innovation Center for Biomedical Engineering, School of Engineering Medicine, Beihang University, Beijing 100191, China

**Keywords:** corneal, biomechanics, bioengineering, ophthalmology

## Abstract

In early corneal examinations, the relationships between the morphological and biomechanical features of the cornea were unclear. Although consistent links have been demonstrated between the two in certain cases, these are not valid in many diseased states. An accurate assessment of the corneal biomechanical properties is essential for understanding the condition of the cornea. Studies on corneal biomechanics *in vivo* suggest that clinical problems such as refractive surgery and ectatic corneal disease are closely related to changes in biomechanical parameters. Current techniques are available to assess the mechanical characteristics of the cornea *in vivo*. Accordingly, various attempts have been expended to obtain the relevant mechanical parameters from different perspectives, using the air-puff method, ultrasound, optical techniques, and finite element analyses. However, a measurement technique that can comprehensively reflect the full mechanical characteristics of the cornea (gold standard) has not yet been developed. We review herein the *in vivo* measurement techniques used to assess corneal biomechanics, and discuss their advantages and limitations to provide a comprehensive introduction to the current state of technical development to support more accurate clinical decisions.

## 1. Introduction

Mechanical characteristics are important features of the cornea. These are dependent on the distribution of collagen fibers in the stroma, the thickest layer of the cornea. The alignment of these internal collagen fibers is directly related to the biomechanical properties and (eventually) to the morphology of the cornea, which further influence its optical properties. Parameters determining corneal biomechanical characteristics can be used to reflect the states of corneal diseases.

A broader and deeper knowledge of corneal biomechanics could promote the evaluation of corneal ectasia patients [1] and facilitate research on treatments such as corneal cross-linking (CXL) [2,3], laser refractive surgery [4,5,6], and corneal transplantation [7]. It could also improve the understanding of the mechanisms of ophthalmic diseases such as refractive abnormalities [8], the effects of various corneal incisions on the corneal structure and function [9], and intraocular pressure determination.

Various parameters are currently used to quantify the corneal biomechanical properties. The longitudinal, shear, and Young’s moduli are the three main classical mechanical parameters that are used. Though different machines gave various terms for the description of corneal biomechanics, they can almost all be related to these three biomechanical features. This review explains the differences between them and their clinical implications. The shear modulus is often used to describe the rigidity of a material and is also referred to as the rigidity modulus. The Young’s modulus, on the other hand, reflects the stiffness of the material [10]. In dynamic loading conditions, the cornea demonstrates viscoelastic behavior; this material property is usually described by the dynamic modulus [11]. The difference between the biomechanical behavior of the cornea during energy storage and energy dissipation is known as “hysteresis” [12]. This property makes accurate measurement of the Young’s modulus of the cornea difficult: the hysteresis during loading and unloading of the intraocular pressure (IOP) causes the Young’s modulus to vary in real time with the IOP [12]. Accurate measurement of the corneal biomechanical properties *in vivo* is of immense importance in clinical practice. At present, various instruments based on different inherent principles are employed to evaluate the cornea based on different biomechanical parameters.

*In vivo* biomechanical measurements of the cornea are still in the nascent stage. The tonometer, which measures corneal rigidity using contact technology, has been available since the early 1900s [13]. Earlier noncontact methods for evaluating corneal mechanical properties were based on the use of air turbulence to cause corneal deformation, enabling noninvasive measurement of the biomechanical properties [14]. Less destructive corneal deformation technologies, such as air puff [15] and ultrasound [16,17], were developed later. Currently, commercial devices are used in clinical settings for measuring the corneal biomechanical properties, but these types of equipment are destructive. Current techniques usually reflect only a few mechanical properties of the cornea, such as longitudinal modulus, shear modulus, or Young’s modulus. There is no device that can comprehensively and accurately measure the various corneal biomechanical parameters. These techniques also make it difficult to reflect the biomechanical properties of different areas of the cornea at the three-dimensional level, often measuring only at the axial level, or showing the average properties of a large area. Moreover, IOP, central corneal thickness (CCT), eye movements, and other factors can cause measurement errors. The development of optical techniques in recent years has made it possible to evaluate corneal mechanical properties in nonperturbative ways [18], and the extensive use of finite element methods in corneal simulations has accelerated these developments [19,20] (Figure 1). These new techniques could provide the spatial distribution of corneal biomechanical features, which is valuable information for clinical use. However, these techniques are still in the early stages of research. There is currently still a lack of a gold-standard technique that can comprehensively quantify the various mechanical parameters of the cornea and describe the corneal biomechanical properties.

This review focuses on various techniques for collecting biomechanical measurements of the cornea *in vivo* and describes their principles of operation, milestones during their development, and application prospects, thus providing an introduction to these existing assessment techniques and their recent developments. We expect this review to inspire new ideas that can promote the development of corneal biomechanical measurement in the future, which will be conducive to the improvement and development of new and more reliable measurements, especially the realization of *in vivo* tracking of corneal stress distribution.

## 2. Perturbation-Based Measurements

Corneal deformation induced by external pressure is a prerequisite for the assessment of corneal mechanical characteristics using perturbation techniques, although the mechanical parameters of interest and modes of analysis may differ within this broad category of techniques. The current direction of development in measurement techniques based on perturbation is from invasive to minimally invasive and noninvasive, with the hope that a noninvasive, noncontact technique will be developed to evaluate corneal biomechanical characteristics.

### 2.1. Ocular Response Analyzer

Early studies conducted to measure the corneal mechanical characteristics usually relied on pneumatic loads to deform the cornea from the inner side. Thus, these studies could only assess the local corneal mechanical properties *in vitro* [21,22,23]. *In vivo* tests implemented by injecting saline into the eyeballs were performed subsequently to determine the pressure–volume relationship and the stiffness of the living cornea [24]. Following this, devices were also developed to measure IOP and corneal rigidity using an *in vivo* contact plunger to flatten the cornea inwards [13]. The ocular response analyzer (ORA) is the first noncontact commercial device to employ this principle to assess corneal stiffness. It estimates the overall biomechanical behavior of the cornea by capturing the corneal deformation as the cornea is subjected to air pressure. The external machine provides an air pressure that varies in magnitude over time to deform the cornea in the inward direction. During this process, two critical points are captured and recorded as P1 and P2 when the central region of the cornea is in a flat shape. The deformation of the cornea is monitored using collimated infrared laser light by detecting the intensity of the reflected infrared signal [14].

Upon data processing, the ORA is able to reveal two parameters: corneal hysteresis (CH) and corneal resistance factor (CRF). The former of these quantifies the corneal viscoelastic properties and the latter is an indicator of the overall corneal resistance, reflecting the corneal elastic characteristics [12,14]. These parameters help to differentiate keratoconus from healthy corneas [25]. Furthermore, the ORA is considered a new IOP measurement technology because biomechanical indicators can be used to obtain the IOP data more accurately, thus providing Goldman-related IOP (IOPg) and corneal-compensated IOP (IOPcc) [14].

Although CH and CRF can reflect the corneal biomechanical properties, it is difficult to distinguish between the biomechanical properties of different areas of the cornea according to their principle and function [26]. The mathematical relationship between CH, CRF, and elastic modulus remains unclear, thus limiting the clinical use of this technique [26]. Additionally, CCT and IOP can influence CRF and CH data. Therefore, the weighted value of the CRF is often used in clinical applications of the ORA [27,28].

To overcome these limitations, several new studies have defined additional parameters [29,30]. These include the hysteretic loop area, which can improve the sensitivity of the ORA by extending the detection range and analyzing the relationship between pressure and displacement throughout the deformation process [30]. The ORA (Reichert Inc., Depew, NY, USA) is now commercially available for clinical use.

### 2.2. Corneal Visualization Scheimpflug Technology

Corneal visualization Scheimpflug technology (Corvis ST) was developed as an improvement of the ORA via the addition of an ultrafast Scheimpflug camera that allows direct monitoring of corneal deformation [31,32,33]. Both the ORA and Corvis ST require similar air perturbations, but the difference is that Corvis ST can maintain a consistent air pressure across different measurements. As a result, Corvis ST can record the process of corneal shape change more accurately, thus enabling more precise assessments [34,35].

Corvis ST reports two parameters, namely, the corneal biomechanical index (CBI) and the total biomechanical index (TBI). Both are new parameters based on a linear regression analysis combining corneal morphology and stiffness parameters, which can improve the detection rate of conical corneas [36]. Other parameters based on Corvis ST that are able to characterize the corneal biomechanical features *in vivo* have recently been proposed [37]. Figure 2 shows an example of keratoconus identified using Corvis ST (Oculus Inc., Wetzlar, Germany), and multiple parameters are quantified.

However, Corvis ST cannot be used to analyze the corneal mechanical behavior in a specific direction. Furthermore, its results are also affected by factors such as the IOP [26]. Repeated experiments using Corvis ST have demonstrated that single measurements do not yield accurate results, and some form of averaging is generally required [38,39]. However, these limitations have not hindered the application of Corvis ST in clinical studies. Corvis ST has been shown to improve the screening accuracy in refractive surgery screening [40]. Corvis ST (Oculus Inc., Wetzlar, Germany) is currently commercially produced and is in clinical use.

### 2.3. Optical Coherence Elastography

Optical coherence tomography (OCT) is a technique used to examine the microscopic structures in biological tissues. The study that first reported it did not target ocular components [41]. Optical coherence elastography (OCE) is an imaging method that combines OCT with external loading to measure the corneal elasticity *in vivo*. It uses tomography for nondestructive estimation of the material properties of the cornea [42]. The evaluation can be performed without a significant increase in the IOP generated by using a pressure source to flatten the cornea. It utilizes the short coherence length of a broadband light source to achieve precise axial and lateral segmentation on a high-scattering medium with high resolution. According to the concept of OCE, the observed tissue deformation can be mathematically modeled and analyzed to approximate the Young’s modulus [43] in order to achieve noninvasiveness. Extensive research has been performed in this area. In addition to the pressure source, a miniature air-puff perturbation method has been proposed which can use a short, high-intensity air puff to cause a local displacement in the cornea; the pulse is propagated in the form of elastic waves, and can thus avoid the measurement errors originating from IOP changes [15,44,45]. The corneal viscoelasticity can also be quantitatively assessed [46].

Some studies have quantified the corneal biomechanical properties after CXL treatment in rabbit eyes using OCE [47], and have assessed the ability of OCE to perform *in vivo* measurements in humans [48]. A number of techniques have been developed based on OCE. Other studies further established a novel noncontact OCE imaging technique that uses interference from external pressure to assess the material properties of the cornea [49]. Air-puff-based corneal deformation is the basis of air-puff OCE [50]. Similarly to ORA, measurements can only be taken on a single axis, so only one-dimensional data can be obtained [50]. Additionally, the mechanical properties of the cornea show heterogeneous variations across the cornea, which makes the specificity of the measurement limited. For the same reason, measurement accuracy can be affected by IOP and CCT [50]. Shear-wave OCE can reflect corneal stiffness by measuring the speed of surface wave propagation through the corneal tissue with an ultrasound device. This makes it possible to take measurements in two or even three dimensions [51]. The results of the measurements still correlate with IOP, CCT, and eye movements [52]. Pulsed laser excitation OCE [53] and acoustic radiation force OCE [54] utilize laser-induced surface acoustic waves and ARF systems as excitation, respectively, which are then monitored in conjunction with OCT techniques. Both allow the Young’s modulus of the cornea to be measured in real time. It is well known that the mechanical properties of the cornea vary with factors such as IOP, and real-time testing can avoid the lagging problems associated with changes in IOP [53]. The techniques are still in the *in vitro* testing phase and *in vivo* measurements have not yet been achieved.

### 2.4. Mechanical Waves

Mechanical waves can be used in perturbation techniques to induce slight tissue displacement. These techniques can reflect the biomechanical properties of the cornea based on propagation analyses of mechanical waves induced in the corneal tissue, thus serving as a new noninvasive technique type used to assess corneal biomechanical properties.

#### 2.4.1. Supersonic Shear-Wave Imaging

Supersonic shear-wave imaging (SSWI) is a novel ultrasound technique that can provide real-time and quantitative mapping of the corneal viscoelasticity in noninvasive conditions to objectively assess soft-tissue stiffness [16,55]. A conventional ultrasonic probe causes corneal displacements, and the ultrasound wave in the tissue evolves into shear waves that propagate and reflect the local elasticity of the cornea based on the speed of their propagation [16]. This technique reflects the viscoelastic index and Young’s modulus of the cornea and is directly applicable to *in vivo* studies.

SSWI can be used to quantify the corneal elastic anisotropy *in vivo* [56], quantitatively estimate the local corneal stiffness, and generate a two-dimensional elastogram of the cornea [16]. SSWI has also shown good evaluation capabilities in CXL treatments [57,58]. This technique has been applied to assess the stiffness of many biological tissues in addition to *in vivo* measurements of corneal biomechanics [59,60,61].

#### 2.4.2. Ultrasound Surface Wave Elastometry

Ultrasound surface wave elastometry (USWE) uses an ultrasound probe to measure the time of wave propagation between two transducers set at a fixed distance; the propagation time can be used to determine the corneal elasticity. This technique reflects the local mechanical properties of the cornea and compares the stiffness values in the central, radial, and other directions [17]. This technique was used for donor tissues in the past [17], but has been successfully applied to *in vivo* examinations of corneas recently [62].

USWE has been applied to the determination of corneal biomechanics in CXL studies [63]. However, this technique has not been adequately studied; as a result, relevant commercial instruments are unavailable.

## 3. Non-Perturbation-Based Measurements

Unlike perturbation-based measurements, non-perturbation-based measurements do not rely on corneal deformation. These techniques usually involve direct corneal biomechanical assessments based on the inherent structural properties of the corneal tissue. The acquisition of the intrinsic corneal tissue parameters can be currently achieved using techniques such as optical or computational simulations.

### 3.1. Brillouin Microscopy

Brillouin microscopy (BM) uses the Brillouin shift between Stokes and anti-Stokes scattering to measure the longitudinal or bulk modulus in order to describe the mechanical compressibility of tissue. BM imaging techniques can be used to obtain volumetric images of the elastic characteristics of the cornea [18]. A low-power, near-infrared laser beam is used to analyze the spectral data of the echo signal and generate a three-dimensional map of the longitudinal modulus of the corneal surface and the corneal thickness.

BM observations can describe the local biomechanical characteristics of the cornea and reflect its three-dimensional spatial heterogeneity. The downsides are the narrow measurement range and the long acquisition time, which limits the large-scale use of BM approaches [64,65]. As for biomechanical properties, BM can be used to assess the longitudinal modulus of the cornea. However, the technology is not currently available for the assessment of Young’s modulus and stress–strain behavior.

Many clinical studies have utilized BM for *in vivo* corneal measurements [65], detection of keratoconus [66], and other applications. Clinical trials are being conducted using BM to demonstrate that corneal CXL treatments can increase corneal stiffness [67]. This technique is expected to become a standard and valid clinical examination technique for the assessment of the biomechanical characteristics of the cornea [68].

### 3.2. Phase-Decorrelation OCT

Phase-decorrelation OCT (PhD-OCT) is a new technique used to characterize corneal biomechanics by combining OCT and BM. Unlike BM, which uses Brillouin scattering, PhD-OCT uses dynamic light scattering of particles in a fluid to obtain the relevant parameters. After scanning, the attenuation constant of the particles, which is related to the viscoelasticity of the material, is calculated using the Fourier transform [69].

PhD-OCT can reflect the corneal spatial heterogeneity in three dimensions [70]. Moreover, PhD-OCT can circumvent most of the disadvantages of corneal biomechanical measurements, such as IOP dependence and extended assessment time, thus making it a promising technique.

### 3.3. Finite Element Method

The finite element method is an *in silico* technique that can mimic real-world physical or mathematical problems by discretizing the complex geometry into a finite number of small volumes with regular shapes, the so-called elements. Regarding the corneal assessment applications, the cornea and its surrounding structures can be represented as a large number of brick or pyramid elements, which are connected through nodes and defined with specific material properties that are ideally derived from patient-specific measurements. The responses to changes in the simulated structure are then obtained by calculating the stiffness matrix. For large deformation analyses, calculation iterations are needed until a converged solution of the entire structure is obtained [71].

The finite element method has been used extensively in several studies of the cornea, and many techniques based on these models have been developed for evaluating corneal biomechanical properties [70,72].

#### Stress–Strain Index Mapping

Stress–strain index mapping is an emerging approach used to assess the biomechanics of corneal materials. This method is based on material model simulations using the finite element method [20] and can be used to analyze the distribution of collagen fibers in the cornea [73,74]. It uses inverse analysis to obtain two-dimensional maps of the corneal stiffness, calculate the material tangential modulus at different IOP levels, and estimate the stress–strain behavior [75]. This approach relies on the proven link between the corneal tissue microstructure and stiffness distribution, and on the concept of the stress–strain index (SSI) proposed in a previous study on Corvis ST [76]. SSI can be used to quantify the corneal Young’s modulus, which is an independent material property that is not influenced by IOP and CCT [75,77]. It has been demonstrated that the SSI parameter gradually decreases as keratoconus progresses [78]. This indicates that the concept of SSI can reflect the biomechanical properties of the cornea and is almost completely independent of corneal morphology and IOP.

The SSI map is a comprehensive analysis of the local SSI values obtained via Corvis ST, using the geometric information and collagen distribution principles to derive the final stress–strain index map. In view of the nonlinear nature of the stress–strain behavior of the cornea, SSI maps can provide more accurate measurements of biomechanical parameters. Before exporting the SSI maps, it is necessary to diagnose the presence and type of corneal ectasia to select the appropriate model for computational analysis. In the case of a healthy cornea, the derived SSI maps show only slight fluctuations in the SSI values across the corneal surface, whereas keratoconus shows great disparities in the SSI value over some areas [75]. Two-dimensional images provide a new tool for the assessment of local changes of corneal stiffness. SSI maps visually quantify the biomechanical characteristics of the corneal surface and provide direct insight into the mechanism of progression of the patient’s keratoconus. However, it should also be noted that the current technique is based on a single model of corneal fiber distribution and does not take into account differences between individual eyes. Moreover, the study only included a fairly small number of keratoconus cases, which makes the final results unrepresentative, and more data need to be included in subsequent research.

The SSI mapping technique has evolved very rapidly and is now in clinical trials. Recent studies have evaluated the progression of keratoconus using SSI maps [79] and demonstrated the technique’s potential for commercial clinical applications.

## 4. Corneal Biomechanics in Clinical Settings

The concern about corneal biomechanics stems directly from clinical problems. The corneal stroma is essentially a thin tissue made up of multiple layers of interwoven fibers, so any disruption of the corneal structure will alter the biomechanical properties of the cornea [80]. One of the most relevant diseases is keratoconus. The most important changes in keratoconus are the reduction in stromal layer thickness and the rupture of the Bowman layer. The collagen fiber content in the cornea is also decreased significantly [81]. This results in a change in the corneal biomechanical properties, which eventually manifests in the morphology of the cornea, with a reduction in the central corneal thickness and a conical shape of the cornea. A study using the OCE technique to assess changes in corneal biomechanical properties *in vitro* in keratoconus and to map the two-dimensional distribution of Young’s modulus showed that the Young’s modulus in the central conical region was much lower than that in healthy corneas [82]. Such comparisons are expected to enable early detection of the changes in biomechanical properties associated with keratoconus and to aid clinical diagnosis. Some systemic connective tissue diseases that manifest in the eye as keratoconus, as well as keratoconus as a complication of refractive surgery, may also be associated with this pathogenesis [83]. To reduce the probability of keratoconus as a complication refractive surgery, most current refractive procedures restrict postoperative corneal stromal bed thickness to a minimum of 250 μm [84]. CXL treatment, which is an effective means of increasing corneal resistance by forming new molecular bonds between the corneal laminae and collagen fibers, is primarily used to treat keratoconus [85]. It has been shown to be effective in enhancing the biomechanical resistance of the cornea, increasing the Young’s modulus in the central conical region, and stiffening the cornea [57].

Outside the field of refraction, glaucoma is also associated with the corneal biomechanical properties. The measurement of IOP is inextricably linked to corneal viscoelasticity [86]. Furthermore, corneal hysteresis (CH) can be an important indicator for the diagnosis of glaucoma and as a prognostic factor for the associated risk: lower CH values may be associated with damage such as lower visual field indices and higher degrees of optic disc defects [86]. It is now thought that CH may be more relevant to changes in glaucoma structure and function than parameters such as CCT. The accurate measurement of CH using ORA has also accelerated the progress and application of related research.

## 5. Conclusions

In recent years, the field of corneal biomechanics has been an area of focus owing to the continuous development of refractive surgery techniques. Table 1 provides an overview of some of the currently available measurement techniques. Several techniques are available for the *in vivo* evaluation of biomechanical characteristics of the cornea, among which ORA (Reichert Inc., Depew, NY, USA) and Corvis ST (Oculus Inc., Wetzlar, Germany) are already in commercial production. Both commercially available devices enable noninvasive, more accurate measurements of biomechanical properties by analyzing the process of corneal large-amplitude deformation. However, the large-amplitude deformation of the cornea shows a nonlinear character, and IOP and CCT can affect the accuracy of the measurement. The analysis method is also based on ultrafast photographic equipment that precludes the possibility of spatially resolved measurements. In response to these problems, measurement techniques such as SSI, USWE, and OCE have been developed. These low-perturbation, low-amplitude techniques enable two-dimensional and even three-dimensional spatial measurements, reflecting variability in the biomechanical properties of different regions of the cornea and improving the safety of detection. It should be noted that these devices still measure based on the perturbation principle, which makes it difficult to avoid sources of interference such as IOP, CCT, eye movements, etc. In view of the unavoidable measurement inaccuracies associated with perturbations, techniques such as BM and SSI mapping have subsequently emerged to assess the biomechanical properties of the cornea, based on optical principles or finite element modeling of the internal structures of the eye. These techniques allow for high-accuracy measurements of the cornea and provide a more comprehensive picture of various corneal biomechanical properties. However, these techniques are still in the early stages of research: BM allows detection in three dimensions, but it has considerable limitation of range and takes a long time to evaluate; while SSI maps are currently only available in two dimensions and need to be modeled separately for multiple diseases and different types of corneas, which is costly. These features limit the use of these devices in clinical applications, such as the early diagnosis of corneal ectasia and incision selection for refractive surgery.

These assessment techniques are being developed and have the potential to enable noncontact, noninvasive clinical examinations, and to provide real-time, three-dimensional, high-resolution maps of the distribution of corneal biomechanical parameters. These measurements will provide more accurate clinical tools for issues such as early screening of keratoconus or risk assessment for refractive surgery, and improve the understanding of corneal diseases.

## Figures and Tables

**Figure 1 bioengineering-10-00120-f001:**
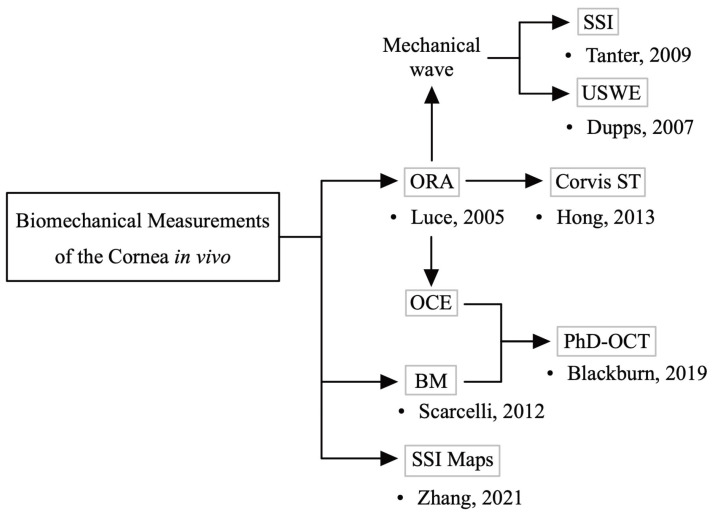
Development of *in vivo* corneal biomechanical measurements. Initial techniques for assessing the corneal mechanical characteristics utilized air puffs (e.g., ocular response analyzer (ORA)) [14]. Later, less destructive corneal deformation techniques, such as ultrasound (mechanical waves) [16,17], were developed. In recent years, new technologies such as brillouin microscopy (BM) [18] have emerged. Some commercial devices (ORA, corneal visualization Scheimpflug technology (Corvis ST)) are now used in clinical settings.

**Figure 2 bioengineering-10-00120-f002:**
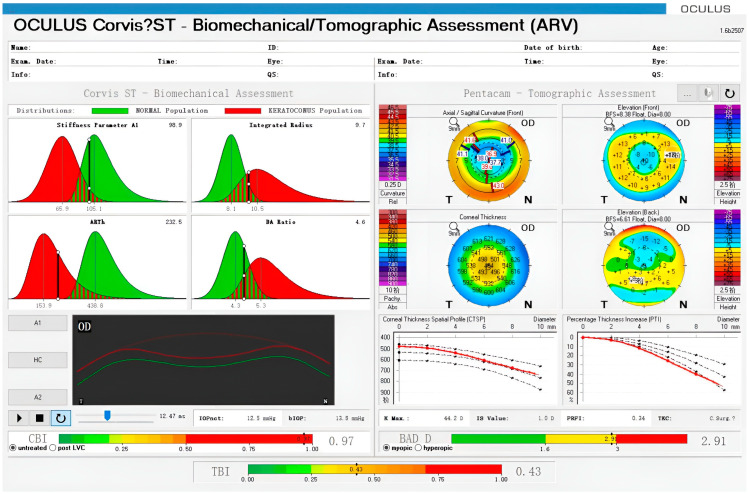
Example of an ectatic eye observed using Corvis ST and Pentacam. Corneal parameters are displayed in the upper left and corneal deformation morphology is displayed in the lower left. Corneal morphological data are shown on the right. The Belin−Ambrósio display (BAD) D is a parameter based on corneal morphology to assess the risk of corneal ectasia. The corneal biomechanical index (CBI) and the total biomechanical index (TBI) are shown at the bottom.

**Table 1 bioengineering-10-00120-t001:** Overview of methods used for *in vivo* corneal biomechanical measurements.

Method	Principle	Advantages	Disadvantages
Ocular Response Analyzer (ORA)	analysis of corneal deformation based on air puffs	the first corneal-mechanics-related instrument that reflects corneal hysteresis versus corneal resistance factor (CRF)	interrupted by IOP and CCT
Corvis ST	direct detection of corneal deformation by using ORA	reflects the corneal biomechanical and tomographic biomechanical indices to accurately record the corneal shape changes	only the uniaxial behavior of the cornea can be analyzed
Optical Coherence Elastography	a tomographic imaging technique for optical coherent elasticity of the cornea	Young’s modulus and viscoelasticity of the cornea can be assessed	difficult to achieve fine measurements *in vivo*
Brillouin Microscopy	analysis of spectral data from echo signals based on Brillouin scattering	the cornea can be described on a three-dimensional level	measurement range limitations and long acquisition times
SSI Mapping	evaluation techniques combining the corneal fiber structure and finite element method	not subject to intraocular pressure (IOP) and central corneal thickness (CCT)	only two-dimensional data are available

## Data Availability

Not applicable.

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
