# Peer review of "In Vivo Biomechanical Measurements of the Cornea"

_bioengineering, 2023, doi:10.3390/bioengineering10010120_

Round 1
Reviewer 1 Report
This is an interesting review of the different methods techniques and devices available and used to better determine the biomechanics and structural characteristics of the cornea. I have no specific comments or queries pertaining to this manuscript.
Author Response
We very much appreciate the thorough review made by the reviewer.
Reviewer 2 Report
The Authors propose a review to provide an update on the recent progress of in vivo biomechanical measurements of the cornea.It summarises the main literature, gives an overview and goes into detail at some parts. Some points need to be clarified. 1. In the Introduction section the Authors should state pro and cons related to the analysis of biomechanical measurements of the cornea. 2. The Section regarding the relationship between the clinical problems and the mechanical characteristics should be included. 3. The Authors should include a Result Section. In this Section they should organize, and explain, the mechanical characteristics changes in corneal diseases.Author Response
We very much appreciate the thorough review and thoughtful comments and suggestions made by the reviewer. Please see the attachment. Thank you.

Reviewer 3 Report
This is a review of currently available techniques/instruments designed to generate a number that can be used to describe the biomechanics of the cornea. Several similar reviews have been published. This paper does not, in my opinion, provide anything new when compared with the available literature. However, I would encourage the authors to consider the following points, re-design, re-write and resubmit this paper.
i) List the various definitions of ‘Biomechanics’. Tell the reader why the term is confusing because the definitions are open to interpretation. Then show the outputs of some of the currently available instruments will be useful depending on the interpretation. This will explain why, some instruments will show some corneas are different from others while others show no difference. Do we need a machine to estimate ‘corneal biomechanics’ or ‘corneal rigidity’? What benefits the clinician: Knowing something about ‘corneal biomechanics’ or ‘corneal rigidity’?
ii) The 1st commercially available procedure designed to measure the rigidity of the cornea in vivo appeared about 80 years ago*. Why ignore this? The instrument is still available, still used and featured in original peer-reviewed papers published since 2000.
iii) Do the biomechanical properties of the cornea depend on the morphological properties of the cornea? A uniform, homogeneous, structureless object has no morphological properties. Yet, it can still have rigidity. Make the body thicker, and it becomes more difficult to bend, stretch or indent it. So, are the biomechanical properties of the cornea dependent upon the morphological properties of the cornea? Are the currently available machines reacting to variations in pachymetry? If not, why not? Argue such points in your review. In your opinion, assuming it’s possible, which technology should provide the ‘gold standard’?
· Friedenwald JS. Contribution to the theory and practice of tonometry. Am J Ophthalmol. 1937;20:985–1024.
Author Response
We very much appreciate the thorough review and thoughtful comments and suggestions made by the reviewer. Please see the attachment. Thank you.

Reviewer 4 Report
This review classifies methods for in vivo biomechanical measurements of the corneal into 2 categories by whether they perturb the cornea or not, and describes the working principles, recent advances, and application prospects of ORA, Corvis ST, OCE, SSWI, USWE, BM, and SSI map. It provides assistance for the future development of in vivo biomechanical measurements of the corneal.
The review summarizes the in vivo biomechanical measurements of the corneal in a standardized, clear and reasonable manner, with a correct understanding of their development and research status, and the advantages and disadvantages of various methods are fully analyzed and discussed, in addition to their future development trends and clinical applications. The author's language is fluent and easy to understand.
Disadvantages: 1. add the analysis of the limitations of OCE; 2. the discussion section can add the aspects in which the in vivo biomechanical measurements of the corneal need to be optimized; 3. improve the clarity of Figure 2; 4. change the paragraph number, such as: "2.1.1" to "2.4.1 ", "2.1.2" to "2.4.2", "3.1.1" to "3.3.1
Author Response

(The authors gave the same response as above.)

Round 2
Reviewer 3 Report
Line 106 & 107: Delete 'in vitro' replace with 'in vivo'
Author Response
Thanks for your comments. We apologize for the mistake that disturbed you. We have modified the revised manuscript.
We have delete 'devices have also been developed to measure IOP and corneal rigidity using an in vitro contact plunger to flatten the cornea inwards' replace with 'devices have also been developed to measure IOP and corneal rigidity using an in vivo contact plunger to flatten the cornea inwards'.